# Atomic mutagenesis in ion channels with engineered stoichiometry

**John D Lueck[1], Adam L Mackey[1], Daniel T Infield[1], Jason D Galpin[1], Jing Li[2], Benoît Roux[2], Christopher A Ahern[1]***

[1]Department of Molecular Physiology and Biophysics, The University of Iowa, Iowa City, United States; [2]Department of Biochemistry and Molecular Biology, University of Chicago, Chicago, United States

**Abstract** C-type inactivation of potassium channels fine-tunes the electrical signaling in excitable cells through an internal timing mechanism that is mediated by a hydrogen bond network in the channels' selectively filter. Previously, we used nonsense suppression to highlight the role of the conserved Trp434-Asp447 indole hydrogen bond in Shaker potassium channels with a non-hydrogen bonding homologue of tryptophan, Ind (*Pless et al., 2013*). Here, molecular dynamics simulations indicate that the Trp434Ind hydrogen bonding partner, Asp447, unexpectedly 'flips out' towards the extracellular environment, allowing water to penetrate the space behind the selectivity filter while simultaneously reducing the local negative electrostatic charge. Additionally, a protein engineering approach is presented whereby split intein sequences are flanked by endoplasmic reticulum retention/retrieval motifs (ERret) are incorporated into the N- or C- termini of Shaker monomers or within sodium channels two-domain fragments. This system enabled stoichiometric control of Shaker monomers and the encoding of multiple amino acids within a channel tetramer.

**\*For correspondence:**
christopher-ahern@uiowa.edu

**Competing interests:** The authors declare that no competing interests exist.

## Introduction

C-type inactivation, also termed 'slow-inactivation' due to its delayed kinetics, in voltage-gated potassium channels describes a phenomenon by which ionic conductance through the channel pore is decreased in the presence of continued depolarizing stimulus (*Choi et al., 1991*; *Hoshi et al., 1991*). In a physiological setting, this mechanism represents a form of activity-dependent macromolecular memory that can alter action-potential firing rates (*Aldrich et al., 1979*; *Roeper et al., 1997*). The basis for C-type inactivation has been investigated with functional, structural and computational approaches, which have produced a consensus mechanism whereby channels become non-conducting due to local perturbations in the selectivity filter (*Yellen et al., 1994*; *Baukrowitz and Yellen, 1995*; *Kiss and Korn, 1998*; *Perozo et al., 1998*; *Roux and MacKinnon, 1999*; *Bernèche and Roux, 2001*; *Cordero-Morales et al., 2007*; *Panyi and Deutsch, 2007*; *Cuello et al., 2010*; *Ostmeyer et al., 2013*; *Thomson et al., 2014*). However, the nature of the conformational change and role of specific residues remain an area of active experimentation and debate (*Devaraneni et al., 2013*; *Hoshi and Armstrong, 2013*).

We previously identified an intra- and inter-subunit hydrogen bond network that controls the rate of entry into the non-conducting pore conformation (*Pless et al., 2013*). Our study employed nonsense suppression in the *Xenopus laevis* oocyte expression system to genetically encode a variety of synthetic noncanonical amino acids (ncAAs). In the context of the *Xenopus laevis* oocyte, the cRNA of the target protein contains an introduced stop codon (often *amber*, TAG) that is subsequently suppressed by a co-injected orthogonal tRNA that has been misacylated in vitro with the ncAA. This approach was used to encode a tryptophan homologue, 2-amino-3-indol-1-yl-propionic acid

(referred to as Ind) that fails to participate in hydrogen bonding at position 434, a site previously identified as being important for channel inactivation. Our original efforts failed to determine the functional role of individual hydrogen bonds with Ind because subunit concatenation in the tetrameric construct resulted in a significant loss of expressing channels, an outcome that prevented the application of nonsense suppression (*Pless et al., 2013*). Reduced channel expression is a common challenge of the nonsense suppression method where amino acid incorporation varies greatly from position to position within a single reading frame and between channel families. Furthermore, given the central role of the Trp434-Asp447 hydrogen bond in the mechanism of C-type inactivation, it was not possible to simultaneously encode Ind within each of the four subunits of the channel because this resulted in a non-conducting phenotype which lacked the relatively high expression levels needed to measure gating currents which originate from the movement of the channel voltage-sensor in the transmembrane electric field (*Perozo et al., 1993*). Thus, we concluded that while one can control the stoichiometry of individual subunit composition in a concatenated construct, their low expression impeded the use of nonsense suppression in combination with this approach. In our original publication, this shortcoming was partially overcome by combining WT cRNA with Ind-tRNA and Trp434TAG cRNA within an individual oocyte. This experimental approach resulted in predicted accelerated, but complex, inactivation kinetics consistent with a mixed population of channels with 1, 2 or 3 Ind434-containing monomers (see Figure 2F, *Pless et al., 2013*, *eLife*). In the present study, we examined the mechanism of Ind-induced inactivation by generating a new approach to control subunit stoichiometry to test individual hydrogen bond contributions in *Shaker* channel C-type inactivation and more generally, for the encoding of multiple ncAA within a single channel or an ion channel complex in the application of nonsense suppression.

## Results

### Molecular dynamic simulations of Ind and the Trp434-Asp447 hydrogen bond

Targeted replacement of Trp434 with the isosteric ncAA Ind was used previously to highlight the general importance of the indole hydrogen bond of this highly conserved Trp residue. The Ind amino acid has seen limited use in structure-function studies for hydrogen bond testing (*Lacroix et al., 2012*; *Pless et al., 2013*, *2014*; *Kim et al., 2015*; *Zhang et al., 2015*) and none thus far in structural or computational studies that would inform on its tolerance once encoded within a protein. For instance, it is not known if the Ind substitution would specifically limit hydrogen bonding at the indole nitrogen atom (the purpose for which it was designed) or could it also produce a new side-chain orientation that itself alters protein function. A computational approach based on molecular dynamics (MD) simulations was therefore employed to examine the local dynamics of the Ind side-chain and its relation to the intra-subunit hydrogen bond accepting side-chain Asp447. These MD simulations reveal key conformational and dynamic differences between wild type (WT) and Trp434Ind of the *Shaker* channel selectivity filter in the conductive state (*Long et al., 2005*) (*Figure 1A*). In WT *Shaker*, a hydrogen bond is formed between Trp434 and Asp447, restraining the two residues close to each other during a large fraction of a 500 ns trajectory (*Figure 1C*). In the absence of the Asp447-Trp434 hydrogen bond in the Trp434Ind *Shaker,* there is an increased distance between these two residues in all of the four subunits (*Figure 1C*), which is associated with a flipping of the Asp447 side-chain toward the extracellular bulk solution (*Figure 1B*). See also Supplemental *Videos 1* and *2* for wildtype W434 and W434Ind channels which depict 150 ns vignettes of the full molecular dynamics simulations.

The breaking of the Asp447-Trp434 hydrogen bond in the Trp434Ind mutant has two major effects in the vicinity of the selectivity filter. First, the formation of a hydrogen bond between Asp447-Trp434 in the wild type blocks the exchange of water molecules between extracellular bulk and the water binding pocket behind the selectivity filter, as shown in the water occupancy map in *Figure 1D*. In the absence of this hydrogen bond in Trp434Ind, the increased spatial separation of Asp447 and Trp434Ind promotes the formation of a continuous connection between the bulk solution and the water molecules in the buried pocket. The easier accessibility of the buried water-binding pocket probably accelerates C-type inactivation of *Shaker*. Second, the flipping of Asp447 leads to an additional effect to the selectivity filter. Considering Asp447 in each subunit carries a negative

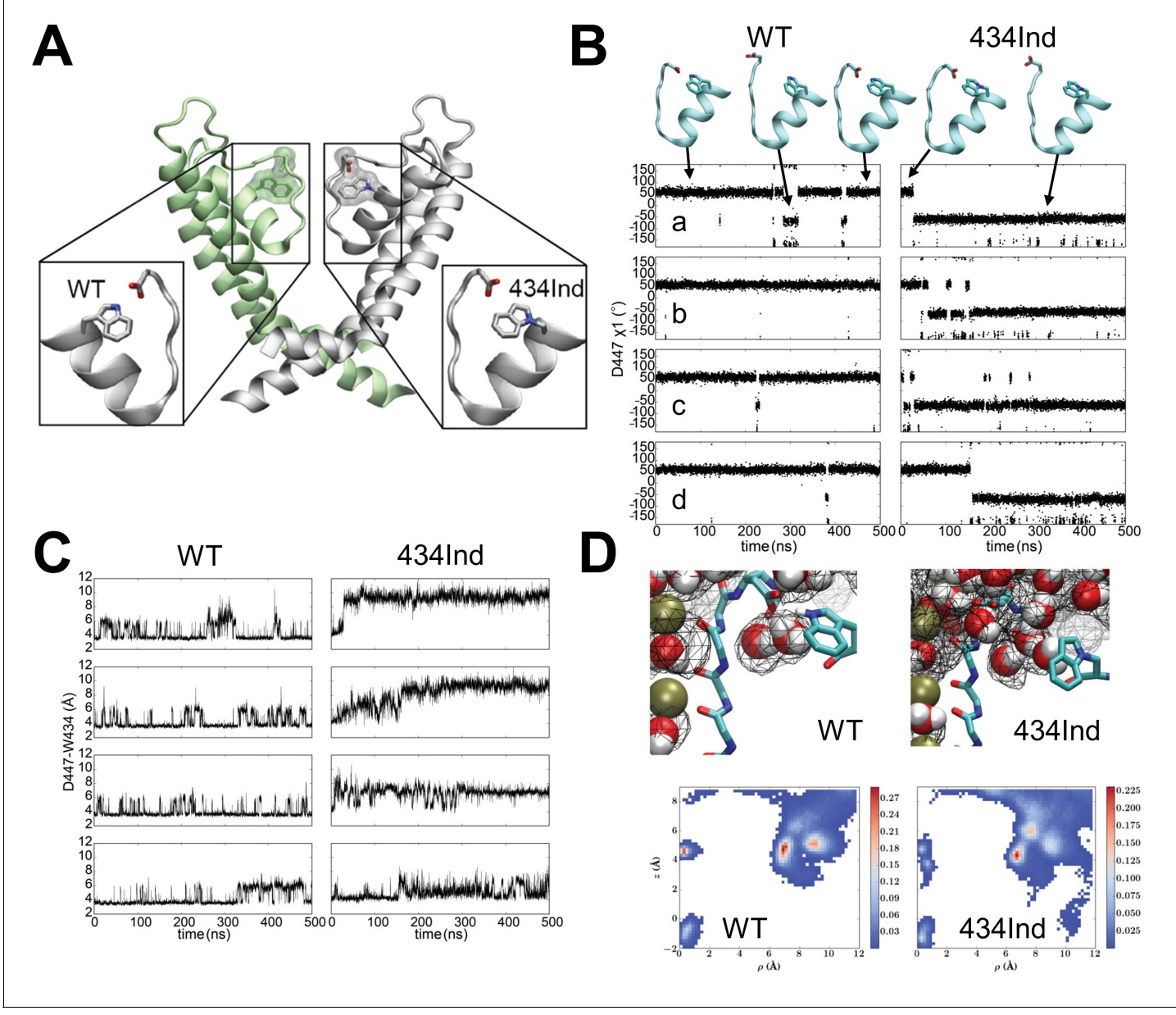

**Figure 1.** Conformational dynamics of wild type and Trp434Ind *Shaker* channel. (**A**) Position of the Trp434 –Asp447 H-bond. (**B**) Time series of dihedral angles χ1 (N–Cα–Cβ–Cγ) of Arp447 for wild type (left) and Trp434Ind (right) demonstrating the flipping of the Arp447 side-chains during the simulations. (**C**) Time series of distance between Asp447 and Trp434 in four subunits characterizing the local conformational dynamics during equilibrium simulations for wild type (left) and Trp434Ind (right) of *Shaker*. The distance is measured between Asp447 (Cγ) and Trp434 (Nε₁) or equivalent atom in the same position in Trp434Ind. (**D**) Top panels, typical conformations for protein and water molecules and occupancy (20%) map of water molecules around selectivity filter of subunit C. Bottom panels, 2D average occupancy map for all four subunits during 500ns MD simulations. The x-axis describes the radius to the center of the selectivity filter, and the y-axis is the z-coordinate of water molecules. In all simulations the selectivity filter in the conductive state. *Videos 1* and *2* indicate 150 ns vignettes of the simulation for the W434-D447 and W434Ind-D447 in the presence shown water molecules. In each case, a subunit of *Shaker* (residues 433 to 447) is shown in new-cartoon representation, and side chains of W434 and D447 are shown in sticks, and water molecules are in VDW representations.

charge, the reorientation of the Asp447 side-chain from facing inside to outside (*Figure 1C*) perturbs the local electrostatic potential at the outer end of the selectivity filter, altering the binding affinity of potassium ions in multiple binding sites, and is likely to shift the relative stability between conductive and nonconductive states.

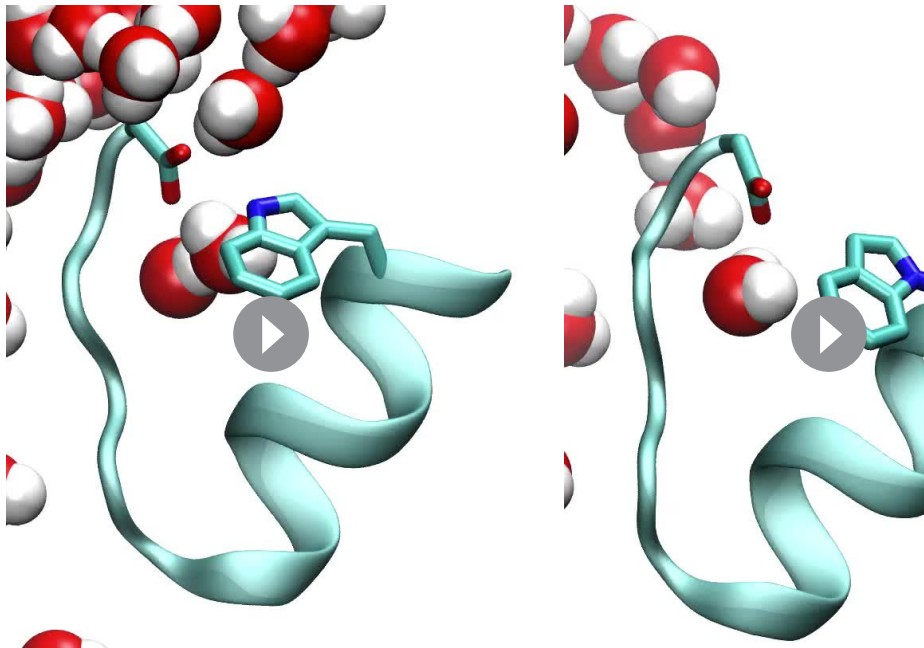

**Video 1.** In the WT a hydrogen bond is formed between Trp434 and Asp447 during a large fraction of MD simulation, which blocks the exchange of water molecules between extracellular bulk and the water binding pocket behind selectivity filter. Upon rare occasion the D447 H-bond is broken spontaneously but quickly reforms.

**Video 2.** In the Trp434Ind *Shaker*, there is a flipping of the Asp447 side-chain toward the extracellular bulk solution, promoting the formation of a continuous connection between the bulk solution and the water molecules in the buried pocket.

## A strategy for defining the stoichiometry of multimeric protein complexes

An engineered protein approach was designed to circumvent issues raised with expressing *Shaker* monomer concatemers: complementary split inteins (*Wu et al., 1998*; *Muralidharan and Muir, 2006*), were incorporated into the N- or C- termini of *Shaker* monomers (*Figure 2A*). Split inteins drive a post-translational event involving precise excision of the intein sequence and formation of a normal peptide bond between the flanking sequences (N- and C-exteins) (*Saleh and Perler, 2006*). We reasoned that *Shaker* monomers would form split intein-dependent dimers, and *Shaker* channels would consist of dimers of dimers. In order to exclude nonconforming *Shaker* monomers from the surface membrane, we encoded endoplasmic reticulum retention/retrieval (ER$_{ret}$) motifs which flank the split inteins (*Figure 2A*) that would be removed following the intein ligation process (*Figure 2B*). We used the Kir6.2 ER$_{ret}$ motif because it works in a variety of eukaryotic cell types, has strong retention properties, and functions independently from its position within a protein (*Zerangue et al., 1999*). Oocytes injected with only one of the monomer cRNAs did not show measurable voltage-dependent K$^+$ selective currents (*Figure 2A*, lower panels). This is presumably due to the retention of the monomers within the ER as a result of the appended ER$_{ret}$ motifs. In contrast, co-expression of *Shaker* monomers with complementary split inteins resulted in robust outward voltage-dependent potassium currents due to the nature of the split intein protein ligation mechanism. The formation of tetrameric dimer-of-dimer complexes with no ER$_{ret}$ motif allows trafficking of *Shaker* channels from the ER to the plasma membrane. The level of expressed potassium current was stable over a large range of injected *Shaker* monomer cRNA concentrations (*Figure 2B*) suggesting a limiting cellular component in channel assembly and/or trafficking. This method allows for control of subunit stoichiometry and the possibility for encoding multiple Ind side-chains within a single channel tetramer by placing the introduced TAG mutation within one (or both) of the two complementary fragments that form a dimer.

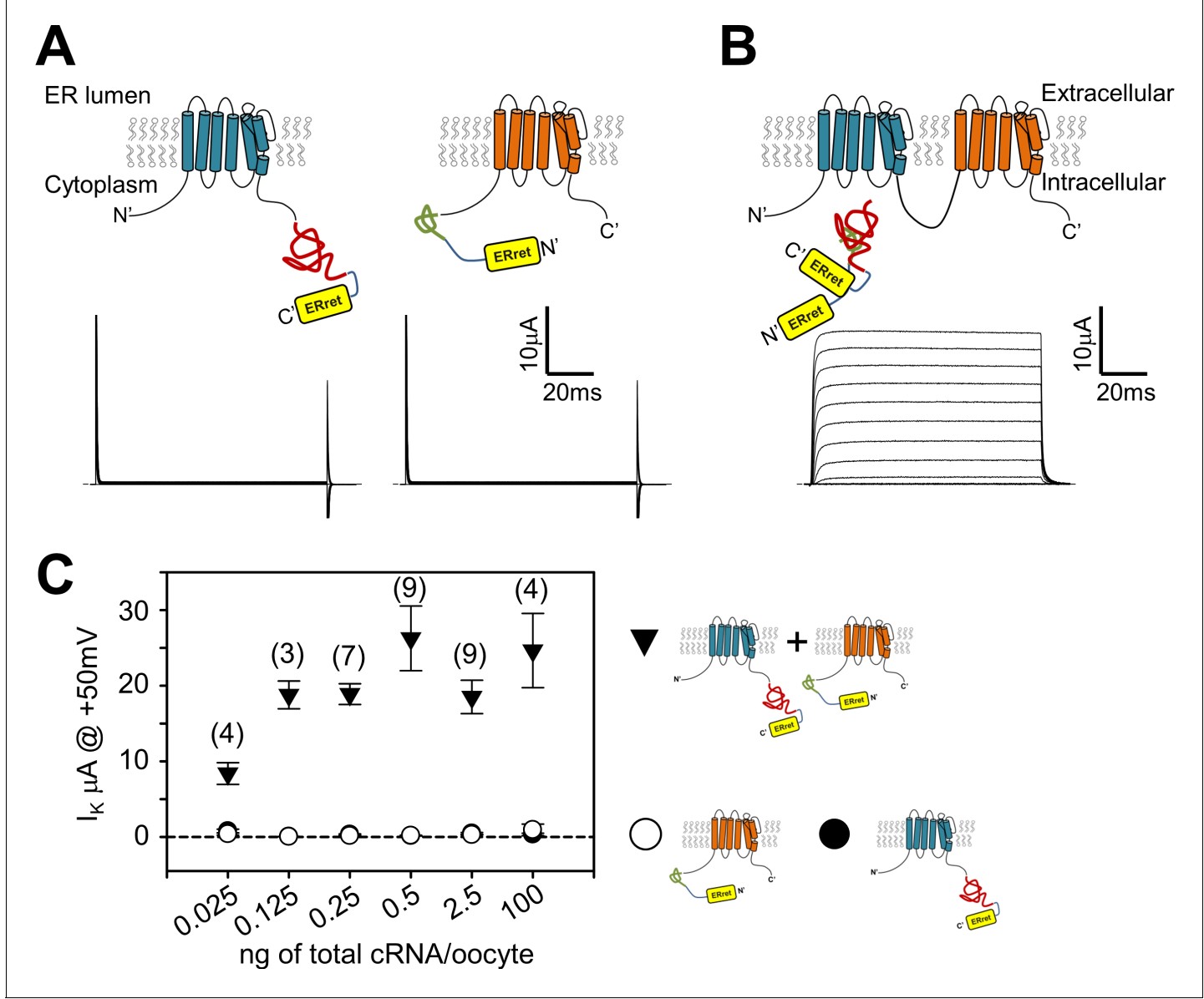

**Figure 2.** Attachment of ER retention/retrieval motifs paired with split inteins results in co-dependent surface membrane expression of *Shaker* channels. (A) Expression of either *Shaker* channel with C-terminal appended Kir6.2 ER retention/retrieval motif (ER$_{ret}$; yellow) with N-intein (red) (A, left) or *Shaker* channel with N-terminal appended Kir6.2 ER retention/retrieval motif (yellow) with C-intein (green) (A, right), resulted in negligible ionic current (A, lower panels).Representative currents in lower panel A, are raw capacitance and ionic currents that are blanked above and below 25 and −5µA, respectively. P/4 subtracted currents are shown in lower panel B to display normal channel kinetics observed following expression of engineered ER$_{ret}$-intein Shaker channels. See Materials and methods for ER-retention/retrieval and intein sequences. (B) Co-expression of *Shaker* N- and C-terminal ER$_{ret}$-intein channels resulted in surface membrane trafficking of *Shaker* channels and robust ionic current (lower panel) measured by TEVC. (C) Injection of increased amounts of C-terminal (filled circles) or N-terminal (open circles) ER$_{ret}$-intein-tagged *Shaker* cRNA resulted in insignificant surface expression of *Shaker* channels as measured with TEVC. Co-injection of increased amounts of N- and C-terminal ER$_{ret}$-intein cRNA resulted in saturated ionic current (filled triangles). Number of observations where two Shaker monomers are expressed are indicated in parentheses above symbols. Average values where only one Shaker monomer was expressed were composed of 2–10 observations.

# Using ER$_{ret}$-Intein protein engineering to encode two ncAA within a *Shaker* channel

In order to test cooperativity of the hydrogen bond at position 434, the ER$_{ret}$-intein approach was used to express *Shaker* channels with two Ind residues at position 434 per channel (*Figure 3*). In experiments where both WT and Trp434TAG complimentary ER$_{ret}$-intein cRNAs and Trp-tRNA were injected, robust WT-like currents were observed, demonstrating successful Trp incorporation (*Figure 3A*, left). Conversely, when Ind-tRNA was expressed within the C-terminal *Shaker* monomer of each dimer, C-type inactivation was significantly accelerated, consistent with the breakage of two of the four hydrogen bonds around the central permeation pathway (*Figure 3A*, right) while voltage dependence of activation was not altered (*Figure 3B*, *Table 1*). At +20 mV, WT *Shaker* channels had an inactivation time constant of 3900 +/- 0.2 milliseconds (first-order exponential) compared to 529 +/-141 and 142 +/-14 milliseconds when one or two hydrogen bonds were broken through Ind incorporation, respectively (*Figure 3C*).

In order to examine the general utility of the ER$_{ret}$-intein sequences, they were constructed within two-domain hemi-channels of the skeletal muscle isoform of the voltage-gated sodium channel Na$_V$1.4. Previously, it has been shown that both sodium and calcium channels are amenable to functional complementation via expression of 'split' hemi-channels consisting of DI-DIII and DIV or DI-II and DIII-DIV, respectively (*Stühmer et al., 1989*; *Ahern et al., 2001*). This result suggests that sodium and calcium channels need not be tethered on a single polypeptide to form a functional tetrameric channel. We therefore reasoned that co-assembly of sodium channel domains I-II and III-IV would likely bring ER$_{ret}$-intein sequences on the C-terminus of DI-II and the N-terminus of DIII-IV in close proximity and promote intein dependent cleavage of the ER$_{ret}$ sequences and formation of a peptide bond between domains II and III (*Figure 4A*). To this end, cRNA for Nav1.4 DI-II and DIII-IV ER$_{ret}$-intein containing hemi-channel fragments were expressed in Xenopus oocyte and subsequent currents were recorded by TEVC. As with the *Shaker* intein constructs, expression of either individual fragment did not result in expressed ionic currents (*Figure 4D*; triangles and squares). However, co-expression of the split Na$_V$1.4 halves resulted in robust inward voltage-gated sodium currents (*Figure 4C– D*, diamonds) similar to WT full-length channels (*Figure 4D*, circles). Notably, full-length sodium channels produced from the ER$_{ret}$-inteins had biophysical properties that were indistinguishable from WT sodium channels, p>0.01, *Table 1*. To examine the general versatility of the approach, we expressed the Na$_V$1.4 hemi-channel construct cDNA by transient transfection into HEK293 cells and examined the ligation efficiency by Western blot using C-terminal HA epitopes that are on both N- and C-channel fragments (*Figure 4B*). In the case of the DI-II-HA construct, this epitope is cleaved out during the in-cell ligation reaction but the full-length channel can still be identified by the remaining C-terminal HA epitope on the DIII-IV protein (see methods for protein sequence and construct design). The Western blot data show that each hemi-channel is robustly expressed, producing bands of ~150 kDa (*Figure 4B*, lane 2) and ~110 kDa (*Figure 4B*, lane 3) for the DI-II-HA and DIII-DIV-HA truncated constructs, respectively. Co-expression of the channel halves resulted in the formation of a ~250 kDa band (lanes 4 and 5) similar to WT Na$_V$1.4 full-length channels (lane 1). The efficiency of protein ligation was similar to that of intein ligated Ca$^{2+}$ channel hemi-channel fragments where also, interestingly, the fragment corresponding to the DIII-IV construct was limiting (*Subramanyam et al., 2013*). To rule out the possibility that the ligation reaction occurred during sample preparation, i.e after the fragments had been solubilized, cell lysates from Na$_V$1.4 DI-II or DIII-IV expressing HEK cells were mixed and then incubated for 10, 30 or 60 min, *Figure 4*, *Figure 1—figure supplement 1*. No higher band corresponding with the full-length Na$_V$1.4 was observed at these time points. Thus, the biochemical data demonstrate that the Na$_V$1.4 DI-II and DIII-IV ER$_{ret}$-intein constructs are competent in HEK cells where they are ligated to produce full-length sodium channels.

Finally, we aimed to see if the split-intein ER$_{ret}$ strategy could assist with nonsense suppression efficiency when attempting to simultaneously suppress two stop sites within a single cRNA transcript. The data in *Figure 3* with *Shaker* ER$_{ret}$-inteins show that this approach can be used to produce tetrameric K$^+$ channel complexes containing two ncAA, but these are within the dimer of dimers K$^+$ channel structural context (i.e. not within the same polypeptide). In contrast, double suppression within the same reading frame poses a particular challenge. Specifically, the ability of an acylated tRNA in the oocyte expression system to suppress stop sites has variable efficiency that is

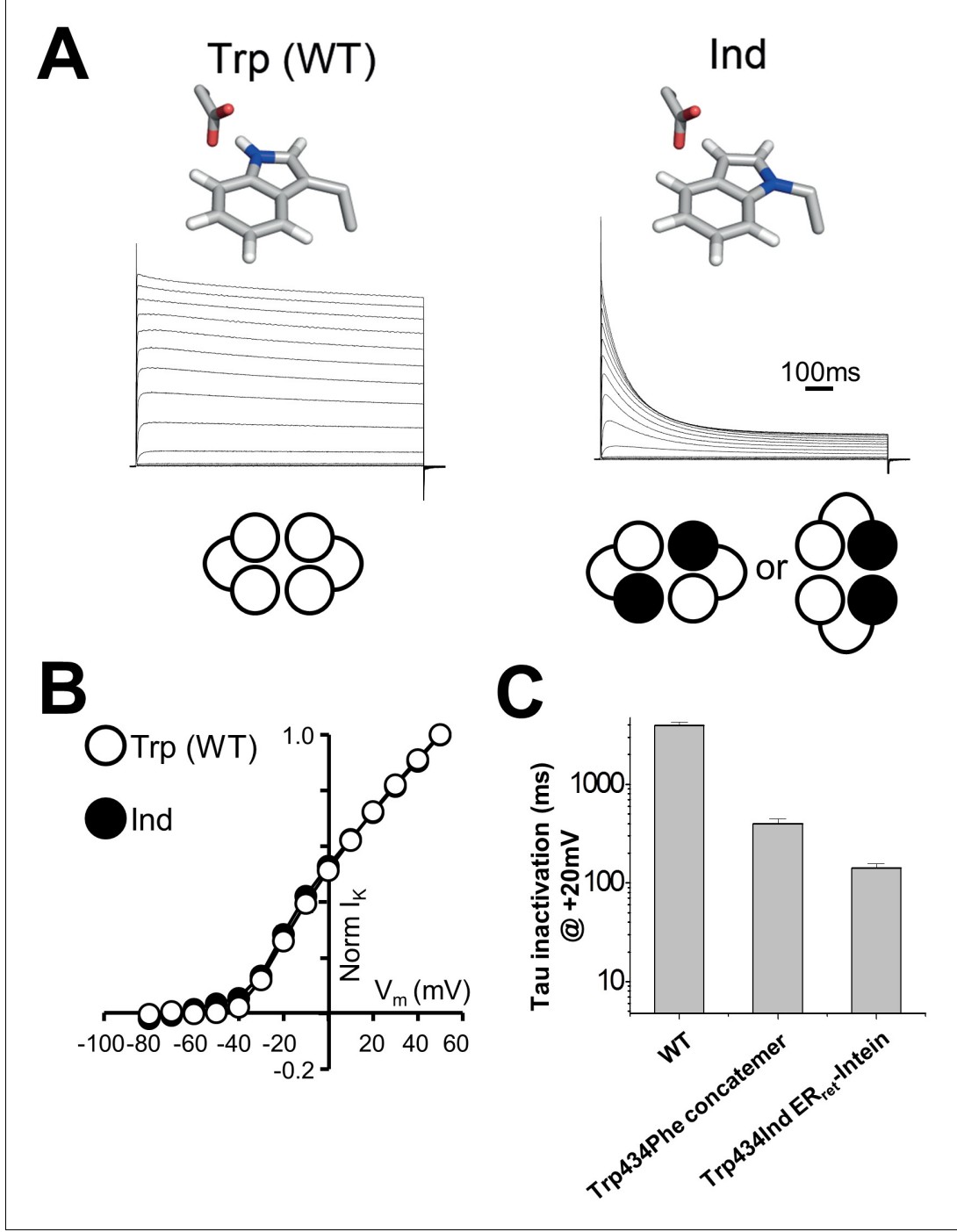

**Figure 3.** Use of $ER_{ret}$-intein tagged *Shaker* monomers allow for stoichiometric expression of Ind. (**A**) Normalized representative currents for Trp434Trp (WT) (left) and Trp434Ind (right). A schematic of the stoichiometry of the incorporated amino acids Trp (white) and Ind (black) is shown below the traces. (**B**) Average voltage dependence of normalized currents for Trp434Trp (WT) (open circles; $n = 5$) and Trp434Ind (filled circles, $n = 9$). (**C**) Average inactivation rates at +20 mV of WT *Shaker*, Trp434Phe (first monomer) four monomer concatemer, and Trp434Ind $ER_{ret}$-intein constructs.

**Table 1.** Shaker and Na$_v$1.4 Channel activation parameters.

| Injected cRNA | V ½ (mV) | K (mV) |
|---|---|---|
| *Shaker Inteins* | | |
| WT-WT (0.25 ng cRNA) | −20.74 ± 2.27 (7) | 11.42 ± 0.55 (7) |
| WT-Trp434Trp (0.25 ng cRNA) | −23.29 ± 1.76 (9) | 9.38 ± 0.85 (9) |
| WT-Trp434Ind (0.25 ng cRNA) | −17.84 ± 2.17 (5) | 7.75 ± 0.74 (5) |
| *Na$_v$1.4* | | |
| Full-length WT Na$_v$1.4 (2.5 ng cRNA) | −25.63 ± 0.71 (5) | 2.70 ± 0.24 (5) |
| Split WT Na$_v$1.4 (2.5 ng cRNA) | −25.97 ± 1.21 (6) | 2.86 ± 0.23 (6) |
| Y401TAG (FH) + WT (BH) + tRNAPhe (25 ng cRNA) | −22.78 ± 0.54 (8) | 3.99 ± 0.13 (8) |
| WT (FH) + F1304TAG (BH) + tRNA(Phe) (25 ng cRNA) | −25.22 ± 1.12 (7) | 4.09 ± 0.19 (7) |
| Y401TAG (FH) + F1304TAG (BH) + tRNA(Phe) (50 ng cRNA) | −24.22 ± 0.55 (7) | 3.78 ± 0.18 (7) |

compounded when two stop sites are present, resulting in the expression of full-length channels that are far below that of WT channels, and often immeasurable using electrophysiological methods. Thus, suppression at the second site down-stream from the first compounds the overall efficiency of producing a full-length peptide. We hypothesized that if the two sites are suppressed independently, within the individual Nav1.4 I-II and III-IV hemi-channels for instance, and then subsequently ligated via the ER$_{ret}$-intein sequences that the reduced expression would not be compounded but only limited by the site with the lower suppression efficiency. To test this possibility directly, stop (TAG) sites were engineered into the DI-II construct at Tyr401 near the outer vestibule of the selectivity filter and within the DIII-IV hemi-channel at Phe1304, a site within the DIII-IV linker. When expressed as hemi-channels, the simultaneous dual suppression with Phe-tRNA at Tyr401 and Phe1304 yielded macroscopic sodium currents with gating similar to wildtype Na$_v$1.4 demonstrating that the split ER$_{ret}$-intein method enables dual suppression, *Table 1*, *Figure 5—figure supplement 1*. The incorporation was specific to encoding of a Phe residue, because co-injecting the Na$_v$1.4 TAG cRNA with a tRNA lacking an acylated amino acid failed to produce sodium current, *Figure 5*.

## Discussion

Taken together, the results presented here advance the specific molecular understanding of C-type inactivation and introduces a novel approach which employs the use ncAAs with split intein protein engineering. MD simulations indicate that the Trp434Ind substitution alters the orientation of the hydrogen bond partner, Asp447. Notably, the 'flipped out' orientation of Asp447 occurs in WT channels, but does so briefly and returns to the hydrogen-bonded state. Thus, the simulations suggest that the Ind substitution increases the likelihood of the 'flipped out' Asp447 conformation, which disrupts the local environment of the selectivity filter. This orientation of Asp447 allows water to fill the space behind the selectivity filter and simultaneously alters the local electrostatics properties, two outcomes that likely impact the occupancy of potassium ions in the filter. One possibility is that the reorientation of Asp447 may represent an event in the mechanism of C-type inactivation as the channel transitions from the open/conductive (O/C) to a C-type inactivated conformation (O/I), see *Panyi and Deutsch, 2007*. Further, the breakage of two versus one of the four Asp447-Trp434 hydrogen bonds further accelerates C-type inactivation, possibly due to the loss of an additional hydrogen bond and ultimately results in a concomitant destabilization of the open pore. However, the data do not inform on the details of movements required to break the hydrogen bonds or the nature of the final non-conducting state, an event that occurs downstream from hydrogen bond breakage.

We previously attempted to understand domain-specific contributions of hydrogen bonding and C-type inactivation with a *Shaker* concatemer with all four subunits tethered into a single peptide (*Yang et al., 1997*). However, while this construct tolerated a single Trp434Phe mutation by site-

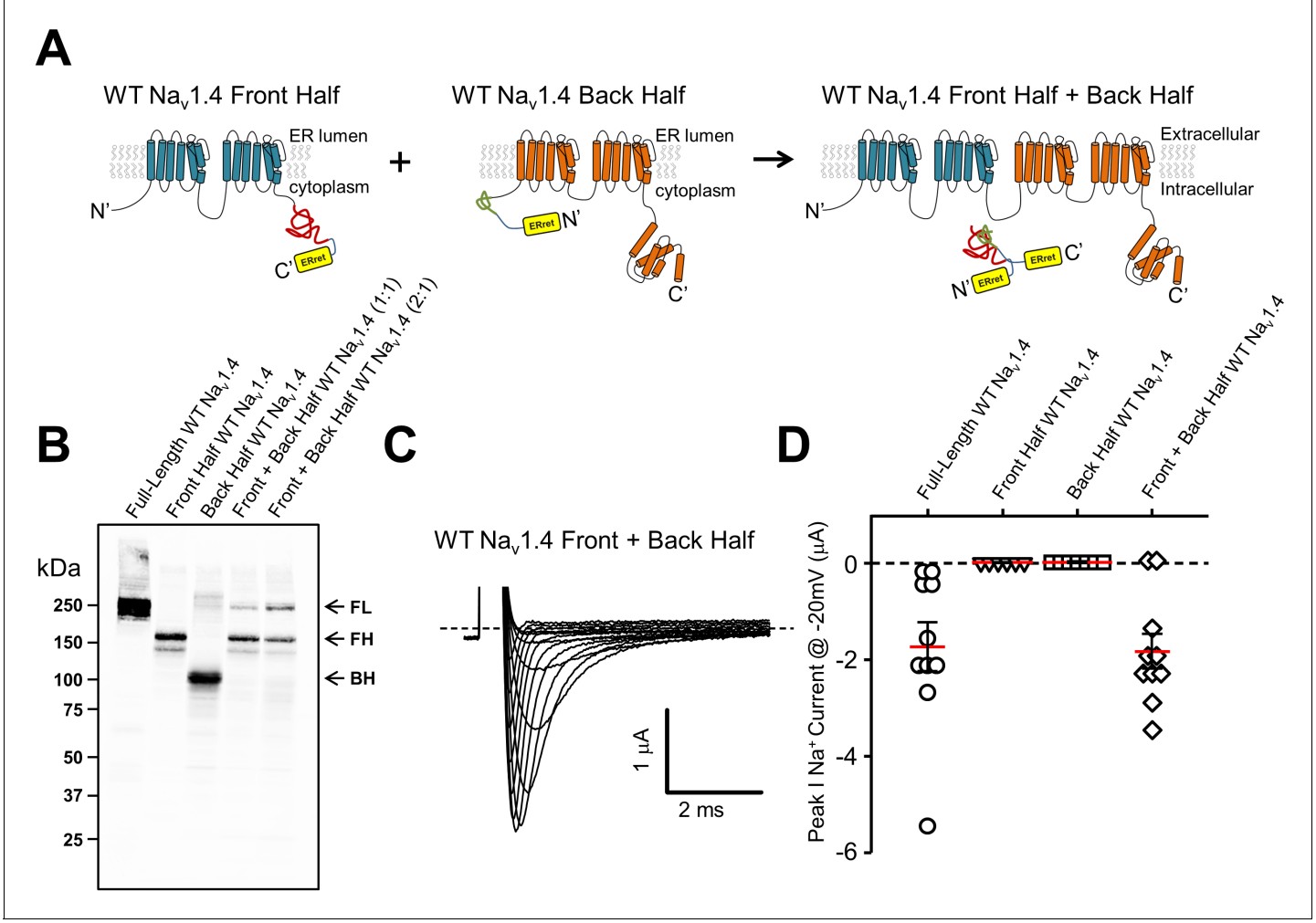

**Figure 4.** ERret-intein construction allows for expression of functional split WT Nav1.4. (**A**) Schematic of Na$_v$1.4 protein with ER$_{ret}$-intein sequence engineered on the C- (blue) and N-terminus (orange) of domains I-II and III-IV, respectively. (**B**) Western blot shows reconstitution and covalent linking of split-intein Na$_v$1.4 proteins following expression in HEK293 cells. Lane 1, full-length WT Na$_v$1.4; Lane 2, front half WT Na$_v$1.4 I-II N-intein; Lane 3, Back half WT Na$_v$1.4 III-IV C-intein; Lane 4 front half WT Na$_v$1.4 I-II N-intein + back half WT Na$_v$1.4 III-IV C-intein expressed at 1:1; Lane 5, front half WT Na$_v$1.4 I-II N-intein + back half WT Na$_v$1.4 III-IV C-intein expressed at 2:1, respectively. (**C**) Representative Na$_v$1.4 currents 12 hr following co-injection of front half WT Na$_v$1.4 I-II N-intein + back half WT Na$_v$1.4 III-IV C-intein. (**D**) Quantification of peak Na$^+$ current @ −20 mV following injection of full-length WT Na$_v$1.4 (circles; n = 10), front half WT Na$_v$1.4 I-II N-intein (triangles; n = 6), back half WT Na$_v$1.4 III-IV C-intein (squares; n = 7) and both front half WT Na$_v$1.4 I-II N-intein + back half WT Na$_v$1.4 III-IV C-intein (squares; n = 10).

The following figure supplement is available for figure 4:

**Figure supplement 1.** Assembly of split-intein Na$_v$1.4 constructs occurs within the cells.

directed mutagenesis (*Pless et. al, 2013*, *eLife*) encoding an ncAA was not feasible as the concatemer displayed a significant decrease in overall expression, a common effect of protein concatenation. Here we describe a protein engineering approach for altering side-chain chemistry prior to concatenation that produces macroscopic potassium currents from *Shaker* channels containing two Ind amino acids. Split intein sequences have been used to ligate a variety of protein fragments (*Shah and Muir, 2014*) and the ER$_{ret}$ motif used in this study has been shown to work in several types of eukaryotic cells including yeast (*Zerangue et al., 1999*). Therefore, this approach could be used to examine other proteins in other cellular environments for studying the effects of dominant negative mutations, heteromers, etc. Western blot data of expressed Nav1.4 ER$_{ret}$-intein hemi-channels demonstrates the effectiveness of the approach outside of the *Xenopus laevis* oocyte

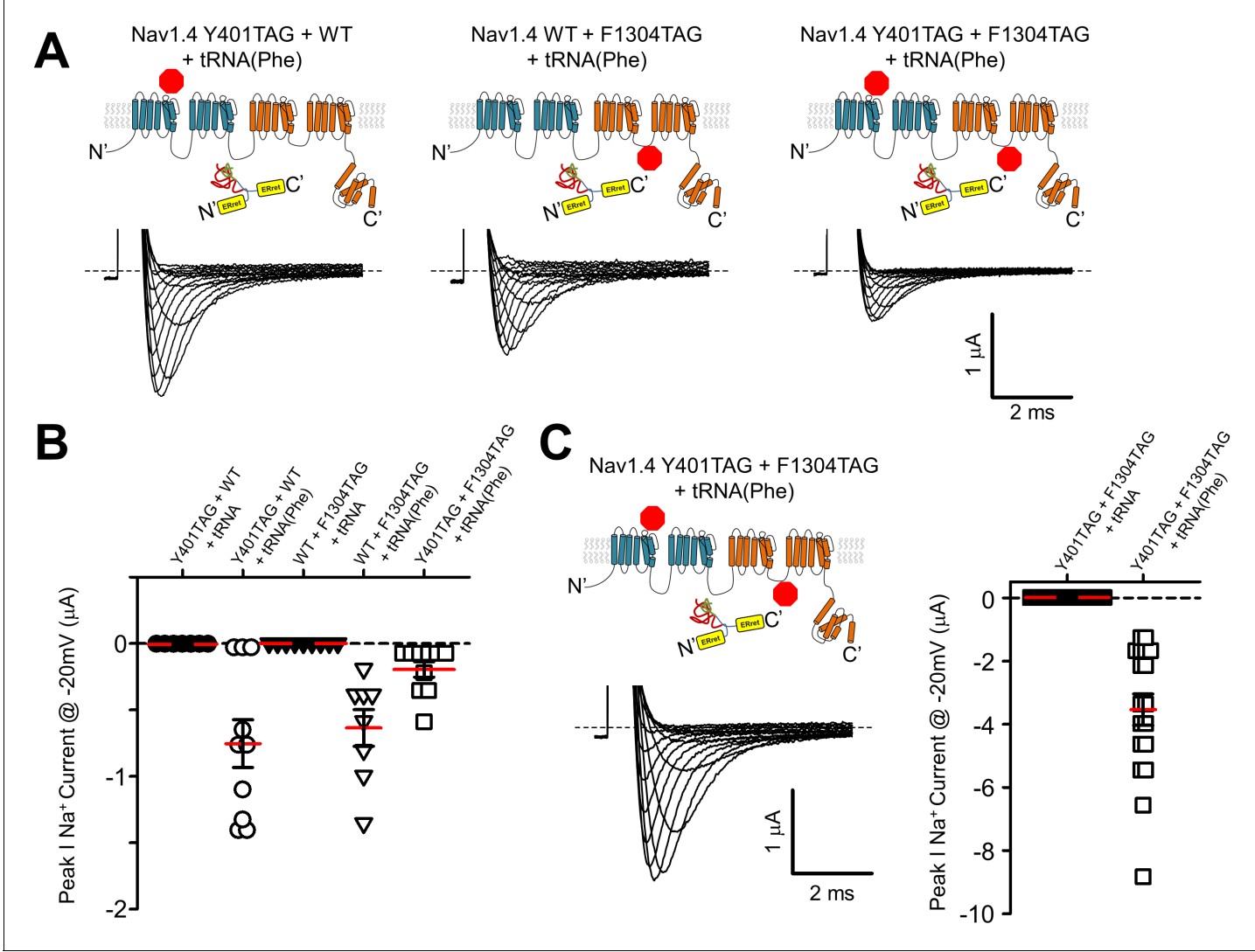

**Figure 5.** ER$_{ret}$-intein construction allows for expression of functional split Na$_v$1.4 with two suppressed codons. (**A**) Co-injection of 25 ng of cRNA encoding Na$_v$1.4 I-II N-intein Tyr401TAG + Na$_v$1.4 III-IV C-intein WT + tRNA(Phe) (left), Na$_v$1.4 I-II N-intein WT + Na$_v$1.4 III-IV C-intein F1304TAG + tRNA (Phe) (center) and Na$_v$1.4 I-II N-intein Tyr401TAG + Na$_v$1.4 III-IV C-intein Phe1304TAG + tRNA(Phe) (right), resulted in robust currents at 24 hr (lower panels). (**B**) Quantification of peak Na$^+$ current @ −20 mV following injection of 25 ng of cRNA of constructs in **A** with tRNA(Phe) (open symbols) and uncharged tRNA (closed symbols). (**C**) Representative Na$_v$1.4 current following co-injection of 50 ng of cRNA encoding Na$_v$1.4 I-II N-intein Y401TAG and Na$_v$1.4 III-IV C-intein Phe1304TAG constructs. (**D**) Quantification of peak Na$^+$ current @ −20 mV following injection of 50 ng of cRNA encoding Na$_v$1.4 I-II N-intein Tyr401TAG and Na$_v$1.4 III-IV C-intein Phe1304TAG (orange) constructs with tRNA(Phe) (open symbols) and uncharged tRNA (closed symbols).

The following figure supplement is available for figure 5:

**Figure supplement 1.** Voltage dependence of Na$_v$1.4 activation is unaltered by engineered split inteins.

expression system and also suggests the ER$_{ret}$-intein can be transferred to other channel types aside from potassium channels. Encoding of multiple ncAA within an individual protein complex remains a significant challenge. Where success is reported, it stems primarily from improving the efficiency of the suppression process and tRNA delivery (*Neumann et al., 2010*; *Wan et al., 2010*; *Johnson et al., 2011*; *Chatterjee et al., 2013*; *Xiao et al., 2013*; *Lammers et al., 2014*). In contrast, the approach described here changes the nature of the dual suppression problem by rescuing each transcript independently prior to protein ligation thus abrogating the loss of expression due to the

compounded efficiency of multiple suppression events. Further, the sodium channel data obtained with dual TAG sites suggests that the approach will be useful for suppression of multiple stop sites, a possibility that will be valuable for encoded FRET, once such probes become available.

## Materials and methods

### Molecular dynamics simulations

Initial coordinates for the molecular dynamic simulations were modeled based on the crystal structure of Kv1.2/Kv2.1 Voltage-gated potassium channel chimera (PDB ID:2R9R). Considering the high sequence identity (87%) of the pore domain between Kv1.2 and Shaker, several non-conserved residues were mutated from Kv1.2 to Shaker, and all of the conserved residue side-chains still maintain the same conformation as the crystal structure. Residues were assigned their standard protonation state at pH7.0. Only the pore domain of the channel was embedded in a bilayer of POPC/POPG (3:1) lipids and solvated in 0.2 M KCl using the web service CHARMM-GUI [1-3] (total number of atoms ~ 44,500). The CHARMM36 force field (*MacKerell et al., 1998*, *2004*; *Klauda et al., 2010*; *Best et al., 2012*) was used for proteins and lipids, and explicit water was described with TIP3P model (*Jorgensen et al., 1983*). For mutant Trp434Ind, the unnatural amino acid was substituted using CHARMM-GUI PDB Reader (*Jo et al., 2014*), and the force field was obtained from CHARMM General force field (CGenFF) (*Vanommeslaeghe et al., 2010*). All-atom simulations were performed in constant NPT conditions at a temperature of 310 K and a constant pressure of 1.0 atm, after initial minimization and equilibrations. All the simulations were performed under periodic boundary conditions with a time step of 2 fs. Throughout the simulations, bond distances involving hydrogen atoms were fixed using the SHAKE algorithm (*Ryckaert et al., 1977*). Short-range non-bonded interactions were calculated using a cutoff distance of 12 Å, and long-range electrostatic interactions were calculated using the particle mesh Ewald (PME) method (*Darden et al., 1993*). Molecular dynamic simulations were carried out using NAMD 2.10 (*Phillips et al., 2005*) or OpenMM 6.2 (*Eastman et al., 2013*). For both wild type and mutant Trp434Ind, MD simulations were performed for 500 ns.

### Molecular biology and nonsense suppression

*Shaker* IR (Inactivation Removed by deletion of amino acids 6–46) cDNA in pBSTA was used as the parent clone for all cDNA plasmids. Plasmids containing the DnaE split inteins (DnaE-n and DnaE-c) were a generous gift from Henry M. Colecraft (Columbia University, New York, NY) and were previoulsy published in *Subramanyam et al., 2013*. Two $ER_{ret}$-intein *Shaker* plasmids were generated. For the first cDNA plasmid, the stop codon was removed and a glycine rich linker (GSAGSAAGSGE-FAS) was added to the c-terminus of the *Shaker* coding region using standard PCR. The DnaE-n sequence (bold) followed by a flexible linker (PINQPANTHERPR), Kir6.2 ER retention/retrieval motif (italics) and canonical KKXX ER retention/retrieval motif (underlined) was generated and ligated into the afformentioned plasmid:

(**CLSYETEILTVEYGLLPIGKIVEKRIECTVYSVDNNGNIYTQPVAQWHDRGEQEVFEYCLEDGSLIRA TKDHKFMTVDGQMLPIDEIFERELDLMRVDNLPN**PINQPANTHERPR*LLDALTLASSRGPLRKRSVA VAKAKPKFSISPDSLS*PR<u>KKFQ</u>stop). For the companion *Shaker* cDNA plasmid, the start codon was removed and a flexible linker (ARSMEAS) was added to the n-terminus of the *Shaker* coding region using standard PCR. The Kir6.2 ER retention/retrieval motif (italics) and DnaE-c sequence (bold) was generated and ligated 5' inframe with the flexible linker using standard molecular biology techniques:

(M*LLDALTLASSRGPLRKRSVAVAKAKPKFSISPDSLS*PR**KIATRKYLGKQNVYDIGVERDHNFALK NGFIASNCFN**). Two $ER_{ret}$-intein rNav1.4 plasmids were generated, Nav1.4 Domain I-II- Intein $ER_{ret}$ and Nav1.4 $ER_{ret}$ Intein III-IV in pcDNA3.1(+) Zeocin (Promega, Madison WI). For the front half Nav1.4 Domain I-II- Intein $ER_{ret}$ plasmid, the DnaE-n intein sequence (bold) was ligated inframe following rNav1.4 Glu985. The intein sequence was followed by tandem human influenza hemagglutinin (*HA*) epitope sequences (underlined) that were used as both flexible linkers and for Western blot analysis. The HA linker was followed by the Kir6.2 ER retention/retrieval motif (italics) and canonical KKXX ER retention/retrieval motif (italics underlined) (**CLSYETEILTVEYGLLPIGKIVEKRIECTVYSVD NNGNIYTQPVAQWHDRGEQEVFEYCLEDGSLIRATKDHKFMTVDGQMLPIDEIFERELDLMRVD NLPN**<u>YPYDVPDYAYPYDVPDY</u>*LLDALTLASSRGPLRKRSVAVAKAKPKFSISPDSLS*PR<u>KKFQ</u>stop). For the

back half Nav1.4 ER$_{ret}$ Intein III-IV plasmid, the ER$_{ret}$ Dna-c intein (bold) and tandem HA epitope (underlined) was ligated 5 'and inframe of rNav1.4 Glu986 (M*LLDALTLASSRGPLRKRSVAVAKAKPKF SISPDSLS*YPYDVPDYAYPYDVPDY**IKIATRKYLGKQNVYDIGVERDHNFALKNGFIASNCFN**).

## In vitro cRNA transcription

WT and Trp434TAG *Shaker* and all Nav1.4 cRNAs were transcribed from a pBSTA vector using the mMessage mMachine T7 Kit (Thermo Fisher Scientific, Waltham, MA). Purification of the cRNA from the transcription reaction was conducted on columns from the RNeasy Mini Kit (Qiagen, Hilden, Germany). Concentration was determined by absorbance measurements at 260 nm and quality was confirmed on a 1% agarose gel (RNase-free).

## tRNA transcription and misacylation

A modified (G73) version of *Tetrahymena thermophila* tRNA, THG73 (*Saks et al., 1996*), lacking the last two nucleotides was transcribed in vitro using CellScript T7-Scribe Standard RNA IVT Kit (CELLSCRIPT). DNA template 20 µg of annealed oligonucleotides coding for the tRNA preceded by a T7 promoter were used (Forward: ATTCGTAATACGACTCACTATAGGTTCTATAGTATAGCGGTTAG TACTGGGGGACTCTAAATCCCTTGACCTGGGTTCGAATCCCAGTAGGACCGC; Reverse: GCGGTCCTACTGGGATTCGAACCCAGGTCAAGGGATTTAGAGTCCCCAGTACTAACCGCTA TACTATAGAACCTATAGTGAGTCGTATTACGAAT; Integrated DNA Technologies, Coralville, IA). The total reaction volume was adjusted to 100 µl and the kit reagents were added in the following amounts: 10 µl of 10X T7-Scribe transcription buffer, 7.5 µl of each nucleotide (100 mM stocks), 10 µl of 100 mM Dithiothreitol, 2.5 µl ScriptGuard RNase Inhibitor, 10 µl T7-Scribe enzyme solution. After the reaction was incubated for 4–5 hr at 37°C, the DNA template was digested with 5 µl DNase (1 U/µl) provided with the kit for 30–60 min. The tRNA was extracted from the reaction with acidic phenol chlorophorm (5:1, pH 4.5) and precipitated with ethanol. The precipitates tRNA was pelleted, washed, dried and resuspended in 100 µl DEPC-treated water and further purified with Chroma Spin-30 columns (Clontech, Mountain View, CA). The procedure yielded roughly 100 µl of ~5 µg/µl tRNA that was stored in aliquots at −80°C. Prior to the ligation reaction of the 73-mer THG73 to the ncAA-pCA conjugate (or just the pCA to yield a complete, nonacylated tRNA), the tRNA was folded in 10 mM HEPES (pH 7.4) by heating at 94°C for 3 min and subsequent gradual cool-down to ~10°C. 25 µg of folded tRNA in 30 µl of 10 mM HEPES (pH 7.4) were mixed with 28 µl of DEPC-treated water, 8 µl of 3 mM ncAA-pCA (or pCA) dimethylsulphoxide stock, 8 µl of 10X T4 RNA Ligase 1 buffer, 1 µl 10 mM ATP and 5 µl of T4 RNA Ligase 1 (New England Biolabs. Ipswich, MA) then incubated at 4°C for 2 hr. The (misacylated) tRNA was extracted from the samples with acidic phenol chlorophorm (5:1, pH 4.5) and precipitated with ethanol. The precipitated tRNA pellets were washed, dried in a Speedvac and stored at −80°C.

## Expression in *Xenopus laevis* oocytes

*Xenopus laevis* oocytes (stage V and VI) were purchased from Ecocyte (Austin, TX). Prior to injection each tRNA pellet was resuspended in 1.5 µl of ice-cold 3 mM sodium acetate and pelleted at 21,000 xg, 4°C for 25 min. For two electrode voltage clamp experiments cRNA amounts are indicated in each figure. After injection, oocytes were kept in OR-3 (50% Leibovitz's medium, 250 mg/l gentamycin, 1 mM L-glutamine, 10 mM HEPES pH = 7.6) at 18°C for ~24–48 hr.

## Two-electrode voltage clamp (TEVC) recordings

TEVC was performed as described previously (*Pless et al., 2011*). In brief, voltage-dependent potassium currents were recorded in in standard Ringer (in mM: 116 NaCl, 2 KCl, 1 MgCl2, 0.5 CaCl2, 5 HEPES, pH 7.4) using an OC-725C voltage clamp amplifier (Warner Instruments, Hamden, CT). Glass microelectrodes backfilled with 3 M KCl had resistances of 0.5–3 MΩ. Data were filtered at 1 kHz and digitized at 10 kHz using a Digidata 1322 A (Molecular Devices, Sunnyvale, CA) controlled by the pClamp 9.2 software. *Shaker* currents were elicited by +10 mV voltage steps from a holding potential of −80 mV to +40 mV and Na$_v$1.4 currents were elicited by +5 mV voltage steps from a holding potential of −100 mV from −80 mV to +40 mV. Clampfit 9.2 software was used for current analysis. Numbers of oocytes, current analysis and statistical significance are indicated in the appropriate figure legends or the main text. All values are presented as mean ± SEM. To determine

statistical significance Student's *t*-test (two-tailed distribution; two-sample equal variance) was performed. The threshold for significance was $p=0.01$.

## Eukaryotic Na$_v$1.4 expression and Western blot

Na$_v$1.4 cDNA plasmids were transfected into HEK293 cells 12–18 hr after plating at 70% confluency using standard calcium-phoshate reagents. 18–24 hr post transfection the HEK293 cells were washed twice with PBS and scraped in PBS supplemented with PMSF (0.1 mM final concentration)and Benzamidine (0.75 μM final concentration) and spun at 4000 *g*. The cell pellet was resuspended in solubilization buffer (50 mM Tris-HCl pH 7.4, 250 mM NaCl, 1% Triton, 0.5 μg/ml Pepstatin, 2.5 μg/ml Aprotinin, 2.5 μg/ml Leupeptin, 0.1 mM PMSF, 0.75 mM Benzamidine) on ice and Vortex Genied for 15mins at 4 C. Non-solubilized protein was removed by centrifugation at 20,000 *g*. Equal cell-lysate was loaded on a 3-15% gradient SDS-page in the presence of 1% 2 mercaptoethanol, separated at 55 V O/N and transferred to 0.45 μM LF PVDF (Bio-Rad, CA, USA). PVDF was immunoblotted using anti-Human influenza hemagglutinin (HA) antibody (1:2000; Biolegend, CA) in 2% NF milk and imaged on LI-COR Odyssey Imaging System (LI-COR, NE, USA).

## Acknowledgements

CAA is supported by NIH/NIGMS (GM106569) and the American Heart Association Established Investigator (A22180002). BR is supported by NIH/NIGMS (GM GM062342). This collaborative work was supported by the Membrane Protein Structural Dynamics Consortium, which is funded by NIH/NIGMS (GM087519). We thank Dr. Stephan Pless comments on the manuscript. The authors declare no conflict of interest.

## Additional information

### Funding

| Funder | Grant reference number | Author |
|---|---|---|
| National Institute of General Medical Sciences | 106569 | Jason D Galpin<br>Christopher A Ahern<br>Daniel T Infield |
| National Institute of General Medical Sciences | 087519 | Jason D Galpin<br>Christopher A Ahern |
| National Institute of General Medical Sciences | 062342 | Jing Li<br>Benoît Roux |
| American Heart Association | A22180002 | Christopher A Ahern |

The funders had no role in study design, data collection and interpretation, or the decision to submit the work for publication.

### Author contributions

JDL, Conception and design, Acquisition of data, Analysis and interpretation of data, Drafting or revising the article, Contributed unpublished essential data or reagents; ALM, JDG, Acquisition of data; DTI, Acquisition of data, Analysis and interpretation of data, Drafting or revising the article; JL, CAA, Conception and design, Acquisition of data, Analysis and interpretation of data, Drafting or revising the article; BR, Conception and design, Analysis and interpretation of data, Drafting or revising the article

### Author ORCIDs

Benoît Roux, http://orcid.org/0000-0002-5254-2712
Christopher A Ahern, http://orcid.org/0000-0002-7975-2744

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
