## [Decision Letter]

Thank you for submitting your article "Atomic mutagenesis in potassium channels with engineered stoichiometry" for consideration by *eLife*. Your article has been reviewed by two peer reviewers, including Baron Chanda (Reviewer #2), and the evaluation has been overseen by Richard Aldrich as the Senior Editor and Reviewing Editor.

The reviewers have discussed the reviews with one another and the Reviewing Editor has drafted this decision to help you prepare a revised submission.

Summary:

The paper of Lueck et al. is a technically sound and very creative report that attempts to address some fundamental questions about the atomic details underlying C-type inactivation gating in K^+^ channels. For many years it has been known that a single substitution at the position 443 of the Shaker K^+^ channel (W434F) yields a protein non-functional, presumably deeply C-type inactivated. C-type inactivation gating is widely accepted to be the functional collapse of the channel's selectivity filter. Structural changes at the channel's selectivity filter have been proposed to produce its structural collapse causing C-type inactivation and ceasing in this way ion conduction. The Lueck et al. paper deal with the identification of molecular interactions that are crucial in determining the conductive conformation of the Shaker K^+^ channel selectivity filter. Particularly interesting to keep the Shaker channel conductive is the highly conserved indole hydrogen bond between the pair Trp434-Asp447. It has been shown before by Dr. Ahern group that the incorporation of a non-hydrogen bonding homologue of tryptophan at position 434 obliterate ion conduction, presumably by constitutively C-type inactivating the Shaker channel, which is similar to the phenotype displayed by the mutation W343F. In this sequel of the previous paper by Pless, Galpin et al. 2013, the authors make two very provocative and significant observations that notably contribute to cement our understanding about C-type inactivation gating in K^+^ channels. First, they propose, based in MD simulation studies, that the Asp447 flips out toward the channel external vestibule and gives space for water molecules to enter a "crack or crevice" behind the channel selectivity filter. This event (the hydration of this crack) and/or a change in the electrostatic potential around the potassium binding sites of the channel's selectivity filter, trigger the functional collapse of latter. Second, the authors in a very innovative way present a new strategy to dissect the contribution of each subunit within the homotetramer channel to the C-type inactivation process. These combined strategies and the ideas derived from this experimental approach I think will be well received among the members of the ion channel community.

Essential revisions:

The following revisions will be required for acceptance:

1) It would have been very powerful and conclusive to perform an SDS PAGE analysis documenting the tetrameric "dimer of dimer complexes" obtained after removal of the ERret motif and putative recovering of trafficking to the oocyte membranes.

2) A more complete statistical analysis is required. Figure 2 is missing a more detailed characterization of the ERret-intein cRNAs constructs to conclude they show "WT-like currents". Please provide the n, statistics and standard deviation of the current expressed by this construct as well as the GV curves to make sure the gating machinery after all these manipulations remain intact. The same applied to Figure 3.

---

## [Author Response]

*[…] The following revisions will be required for acceptance:*

*1) It would have been very powerful and conclusive to perform an SDS PAGE analysis documenting the tetrameric "dimer of dimer complexes" obtained after removal of the ERret motif and putative recovering of trafficking to the oocyte membranes.*

We agree completely with this sentiment. The resubmitted manuscript now contains new Western blot data demonstrating the high fidelity of in-cell protein ligation using newly designed sodium hemi-channels (Figure 4). The reason for using sodium channels was two-fold. First, to further demonstrate the versatility of the approach in terms of channel type, second, the expression system used for these experiments was HEK cells as opposed to *Xenopus oocytes*. Thus the data support the notion that the ERret/intein sequences are generally tolerated between channel and cell types.

2) A more complete statistical analysis is required. Figure 2 is missing a more detailed characterization of the ERret-intein cRNAs constructs to conclude they show "WT-like currents". Please provide the n, statistics and standard deviation of the current expressed by this construct as well as the G/V curves to make sure the gating machinery after all these manipulations remain intact. The same applied to Figure 3.

We now include a table (Table 1) summarizing the potassium and sodium channel electrophysiological data (GV curves and Boltzmann fits) with pairwise statistical analysis and sample sizes.